# Mechanisms and Effect of Coptidis Rhizoma on Obesity-Induced Inflammation: In Silico and In Vivo Approaches

**DOI:** 10.3390/ijms22158075

**Published:** 2021-07-28

**Authors:** Oh-Jun Kwon, Ji-Won Noh, Byung-Cheol Lee

**Affiliations:** Department of Clinical Korean Medicine, Graduate School, Kyung Hee University, 26 Kyungheedae-ro, Dongdaemun-gu, Seoul 02447, Korea; kwonjob@naver.com (O.-J.K.); oiwon1002@naver.com (J.-W.N.)

**Keywords:** Coptidis Rhizoma, obesity, adipose tissue, macrophage, inflammation

## Abstract

Obesity is characterized as a chronic, low-grade inflammation state accompanied by the infiltration of immune cells into adipose tissue and higher levels of inflammatory cytokines and chemokines. This study aimed to investigate the mechanisms and effects of Coptidis Rhizoma (CR) on obesity and its associated inflammation. First, we applied a network pharmacology strategy to search the target genes and pathways regulated by CR in obesity. Next, we performed in vivo experiments to confirm the antiobesity and anti-inflammatory effects of CR. Mice were assigned to five groups: normal chow (NC), control (high-fat diet (HFD)), HFD + CR 200 mg/kg, HFD + CR 400 mg/kg, and HFD + metformin 200 mg/kg. After 16 weeks of the experimental period, CR administration significantly reduced the weight of the body, epididymal fat, and liver; it also decreased insulin resistance, as well as the area under the curve of glucose in the oral glucose tolerance test and triglyceride in the oral fat tolerance test. We observed a decrease in adipose tissue macrophages (ATMs) and inflammatory M1 ATMs, as well as an increase in anti-inflammatory M2 ATMs. Gene expression levels of inflammatory cytokines and chemokines, including tumor necrosis factor-α, F4/80, and C-C motif chemokine (CCL)-2, CCL4, and CCL5, were suppressed in adipose tissue in the CR groups than levels in the control group. Additionally, histological analyses suggested decreased fat accumulation in the epididymal fat pad and liver in the CR groups than that in the control group. Taken together, these results suggest that CR has a therapeutic effect on obesity-induced inflammation, and it functions through the inhibition of macrophage-mediated inflammation in adipose tissue.

## 1. Introduction

The pandemic of obesity is a substantial public health crisis and has caused a dramatic rise in metabolic diseases. It is now widely accepted that several obesity-related diseases, such as dyslipidemia, steatohepatitis, type 2 diabetes mellitus, and cardiovascular diseases, share common characteristics of chronic, low-grade inflammation [1]. The mechanism linking inflammation and metabolic syndromes remains unclear [2], but previous studies have suggested that the infiltration and activation of immune cells, including dendritic cells, lymphocytes, monocytes, and macrophages, in adipose tissue contributes to the pro-inflammatory state and pathogenesis of insulin resistance [3]. Therefore, understanding the inflammatory response in obesity may aid in the discovery of novel therapies for various metabolic disorders.

Coptidis Rhizoma (CR), also called Huang-Lian in Chinese, is the rhizome of *Coptis chinensis* Franch that belongs to the Ranunculaceae family. CR has various functions, such as removing dampness, eliminating toxins, and clearing excess heat according to oriental medicinal theory [4]. It has been used to treat numerous inflammatory diseases and metabolic imbalances for centuries. CR has been found to ameliorate insulin resistance, glucose metabolism, and obesity in multiple experimental studies [5,6,7]. However, most previous studies have focused on the actions of specific alkaloids of CR such as berberine, even though CR is prescribed as a whole herb in oriental medicine. Hence, the goal of this study is to screen potential mechanisms of CR itself on obesity and investigate its effect by evaluating phenotypic changes, ATM populations, and the gene expression of inflammatory markers. Metformin, which is the first-line medication for type 2 diabetes, was used as a positive control, as it has been reported to improve glucose intolerance and reduce body weight gain in high-fat diet-fed mice [8]. This study is an integrated in silico and in vivo study to determine the effects of CR on obesity-induced inflammation and its related target genes and pathways.

## 2. Results

### 2.1. Network Pharmacology-Based Analysis

#### 2.1.1. Analysis of the Ingredient–Target Network

We identified 12 bioactive ingredients of CR (Appendix A) and their 428 target genes (Appendix A). We also collected 2861 obesity-related target genes (Appendix A) and obtained 217 overlapping genes (Appendix A) between target genes of the 12 ingredients of CR and those of obesity (Figure 1a). The CR ingredient–obesity target interaction network had 229 nodes, of which 217 represented the target of the action, and 486 edges indicated the relationship between the nodes (Figure 1b).

#### 2.1.2. Analysis of the Protein–Protein Interaction (PPI)

We observed 217 nodes with 859 edges in the network (Appendix A). Secondly, we extracted the top 10 hub genes based on the degree of connectivity (Figure 1c). These 10 hub genes, which are described in detail in the discussion, are mainly associated with macrophage activation in obesity.

#### 2.1.3. Functional Enrichment Analysis

The Gene Ontology (GO) enrichment analysis helped identify 146 molecular function (MF) terms, 77 cellular component (CC) terms, and 615 biological process (BP) terms. We extracted the top 10 significantly enriched terms from the three categories according to the number of genes involved and combined them to construct a dot plot showing 30 terms (Figure 2a). In the MF category, molecular-level activities performed by target proteins include protein binding, ATP binding, and protein kinase activity. In the CC category, locations on cellular structures in which a target protein performs a function are described, such as the plasma membrane, cytosol, and cytoplasm. In the BP category, complex biological processes are described, such as signal transduction, protein phosphorylation, and the negative regulation of apoptotic processes. We also performed Kyoto Encyclopedia of Genes and Genomes (KEGG) pathway enrichment analysis to further elucidate the mechanism of CR in obesity. We identified the top 30 pathways sorted by gene count (Figure 2b). The enriched genes were linked to insulin-related pathways, such as the insulin signaling pathway and insulin resistance pathway, and several inflammation-related signaling pathways, such as the phosphatidylinositol3-kinase (PI3K)/protein kinase B (AKT) signaling pathway, cyclic adenosine monophosphate (cAMP) signaling pathway, chemokine signaling pathway, mitogen-activated protein kinase (MAPK) signaling pathway, tumor necrosis factor (TNF) signaling pathway, and toll-like receptor (TLR) signaling pathway. These findings indicate that bioactive ingredients in CR may act on multiple pathways to reduce inflammation and ameliorate insulin resistance in obesity. Among those obtained pathways, we noticed the TNF signaling pathway and chemokine signaling pathway, which are closely related to macrophage activation in adipose tissue. These results provided clues about the mechanisms of CR in treating obesity and the design of in vivo experiments to confirm its effects. Detailed GO and KEGG pathway data are presented in Appendix A.

#### 2.1.4. Molecular Docking

Molecular docking was the last in silico approach before the initiation of the in vivo experiment. We aimed to simulate whether CR could trigger sufficient effects on macrophages and their released TNF based on our earlier in silico findings. We discovered high-affinity binding between the three representative bioactive ingredients (berberine, palmatine, and coptisine) of CR and proinflammatory M1 macrophages (Figure 3a–c), anti-inflammatory M2 macrophages (Figure 3d,f), and TNF (Figure 3g,i).

### 2.2. Evaluation of Antiobesity Effects of CR In Vivo

Through in silico screening, we found that CR might act on macrophage activation and modulate inflammation in treating obesity. Thus, we conducted an in vivo experiment to confirm its anti-inflammatory effects and discovered that CR regulated the macrophage infiltration and gene expression of inflammatory markers. We pointed out macrophage-mediated inflammation as the key factor connecting in silico and in vivo approaches.

#### 2.2.1. Effects on Weight-Related Outcomes

We observed significant weight gain in the whole body, liver, and epididymal fat pad in the control group than that in the NC group. Change in body weight in the control group and CR groups showed a significant difference despite similar caloric intake (Figure 4a,b). Body weight changes over time are shown as a graph (Appendix A). We identified that CR 200 administration improved every weight-related outcome. In the CR 400 group, only the weight of the epididymal fat pad was significantly reduced (Figure 4c,d).

#### 2.2.2. Effects on Glucose Metabolism and Insulin Resistance

According to the OGTT, the control group displayed a significantly greater AUC than the NC group, and both the CR 200 and CR 400 groups showed a significantly smaller AUC than the control group (Figure 4f,g). To evaluate insulin resistance, fasting insulin levels, and HOMA-IR were measured. We observed elevated insulin levels in the control group than those in the NC group and significantly lower insulin levels in both the CR 200 and CR 400 groups than those in the control group. A similar pattern was observed for HOMA-IR (Figure 4h,i).

#### 2.2.3. Effects on Lipid Metabolism

CR effectively decreased the total cholesterol and LDL cholesterol, and increased HDL cholesterol (Figure 5a). Free fatty acid levels were significantly decreased only in the CR 200 group than those in other groups (Figure 5e). Phospholipid and triglyceride levels were not significantly decreased in both the CR 200 and CR 400 groups (Figure 5c,d). The OFTT results and AUC calculation showed a significant decrease in both the CR 200 and CR 400 groups than in the control group (Figure 5b,f).

#### 2.2.4. Effects on the Liver and Kidney Function

We observed significantly elevated liver enzyme levels in the control group than those in the other groups. The CR 200 group exhibited lower AST and ALT activity than the control group (Appendix A). The CR 400 and MET groups showed protective effects in AST, but not ALT. Moreover, we observed a significant elevation in the serum creatinine levels in the control group than those in the NC group. A significant decrease in serum creatinine levels was observed only in the MET group than that in the control group (Appendix A).

#### 2.2.5. Effects on Adipose Tissue Macrophages (ATMs)

We observed a significantly increased percentage of total ATMs in the control group than that in the NC group. Both the CR 200 and CR 400 groups showed a significantly lower percentage of ATMs than the control group (Figure 6a,b). A similar pattern was observed in the percentage of CD11c+ ATMs that were closely related to inflammatory actions and also referred to as M1 macrophages. The percentage was significantly higher in the control group than that in the NC group. Both the CR 200 and CR 400 groups had a lower percentage of CD11c+ M1 ATMs than those in the control group (Figure 6c). The percentage of CD206+ ATMs, which are related to anti-inflammatory actions and also referred to as M2 macrophages, was significantly lower in the control group than in the NC group. The CR 200 and CR 400 groups had a significantly higher percentage of CD206+ M2 ATMs than the control group (Figure 6d).

#### 2.2.6. Effects on Inflammatory Gene Expression in Adipose Tissue

The control group showed a significant elevation in every gene expression (TNF-α, F4/80, CCL2, CCL4, CCL5, and CXCR4) than the NC group. Administration of CR 200 significantly lowered the expression of TNF-α, F4/80, CCL2, CCL4, and CCL5. Administration of CR 400 significantly lowered the expression of TNF-α, F4/80, and CCL5. The MET group showed a downregulated expression of TNF-α and CCL2 than the control group (Figure 6e–j).

#### 2.2.7. Effects on the Size of Lipid Droplets in Hepatic Tissue and Adipocytes in Adipose Tissue

We observed that the control group showed a prominent increase in the size of lipid droplets in the liver and adipocytes in the epididymal fat pad than the NC group. Both the CR 200 and CR 400 groups showed significantly downsized lipid droplets and adipocytes than the HFD group (Figure 7a). The fat area in the liver and epididymal fat pad was significantly decreased in both the CR groups than that in the control group (Figure 7b,c).

## 3. Discussion

The study aimed to explore the mechanisms of CR against obesity through network pharmacology and verify its effects using in vivo experiments. First, we conducted network analysis to identify key ingredient–target protein interactions and hub pathways in treating obesity. Second, based on the findings from network pharmacology, we performed experiments focusing on adipose tissue inflammation by observing macrophage modulation and inflammatory gene expression. The findings indicate that the oral administration of CR suppressed macrophage infiltration and inflammatory gene expression in adipose tissue, which led to the identification of multiple ameliorated metabolic markers. This is the first study to elucidate the cellular, molecular, and genetic mechanisms of CR in obesity and its associated inflammation.

Based on the PPI analysis of 217 overlapping genes, we identified the top 10 target proteins with a high degree of connectivity that may represent critical molecular targets mediating the antiobesity effects of CR. *PIK3R1* and *PIK3CA* encode subunits of phosphatidylinositol 3-kinase (PI3K); however, their actions are incompatible because *PIK3CA* encodes the catalytic subunit of PI3K, while *PIK3R1* encodes the regulatory subunit. McCurdy et al. reported that the attenuated expression of *PIK3R1* reduced inflammation and macrophage accumulation in adipose tissue in obesity [9]. PI3K phosphorylates AKT, and the PI3K/AKT pathway regulates macrophage activation by converging metabolic and inflammatory signals [10]. AKT1 is a member of the protein kinase B that regulates cell survival, angiogenesis, proliferation, and metabolism, and AKT1-deficient macrophages in mice lead to the increased secretion of proinflammatory cytokines and infiltration of macrophages [11]. Amyloid precursor protein (APP) has also been reported to increase in ATMs in obese mice [12], and the depletion of APP modulates diet-induced weight gain and reduces macrophage cytokine secretion [13]. Formyl peptide receptor 2 (FPR2) also plays a significant role in HFD-induced obesity, and the ablation of FPR2 reduces inflammation by inhibiting macrophage infiltration and M1 polarization in adipose tissue [14]. Melanin-concentrating hormone receptor 1 (MCHR1), which is widely distributed in the brain, manipulates feeding behavior, and the depletion of MCHR1 ameliorates glucose metabolism and reduces adiposity with improved thermoregulation against cold [15]. Steroid receptor coactivator (SRC) is a tyrosine protein kinase that plays various roles in macrophage-mediated inflammatory responses and cytokine release [16]. Mitogen-activated protein kinases (MAPKs) produce proinflammatory cytokines, chemokines, and matrix metalloproteinases (MMPs) and recruit M1 macrophages in response to diverse stimuli [17]. Previous reports indicated that MAPK1 contributes to the elevated secretion of monocyte chemoattractant protein-1 (MCP-1) [18], and MAPK3 is associated with enhanced mRNA levels of inflammatory markers, such as CCL2, IL1β, and TNF-α in obesity [19]. Signal transducer and activator of transcription 3 (STAT3), a transcription factor associated with immune function and energy metabolism [20], promotes inflammation in obese adipose tissue by modulating T cells, and knockout of STAT3 in T cells reduces macrophage accumulation and restores the M2 phenotype in adipose tissue [21]. Taken together, these 10 genes are related to obesity-induced inflammation and primarily function to modulate macrophages.

To further explore the diverse mechanisms of CR against obesity, we performed functional enrichment analysis that indicated the inflammatory response in the GO term and TNF signaling pathway and chemokine signaling pathway in the KEGG terms. Genes involved in these pathways were mainly linked to inflammation caused by the infiltration of immune cells, especially macrophages, and were associated with insulin resistance. The link between insulin resistance and obesity-associated inflammation has been reported in terms of TNF-α, which is a potent and critical multifunctional proinflammatory cytokine [22]. It has been shown that enhanced TNF-α levels lead to insulin resistance and that the inhibition of TNF-α ameliorates obesity-induced insulin resistance [23]. Thus, the TNF signaling pathway, which plays a substantial role in cell differentiation, apoptosis, the modulation of immune responses, and the induction of inflammation, could be a critical target of CR in treating obesity. We also focused on the chemokine signaling pathway involved in pathological conditions, such as rheumatoid arthritis, multiple sclerosis, atherosclerosis, diabetes, and obesity. Chemokines are mediators of innate immune cell trafficking in response to inflammatory cues and are also central cellular players in adaptive immune responses [24]. Elevated chemokine expression or unregulated chemokine signaling is associated with persistent or excessive inflammation, which is characteristic of chronic inflammatory and autoimmune diseases. Therefore, chemokines, receptors, and their signaling pathways may be therapeutic targets in obesity. These findings indicate that the modulation of macrophages, inflammatory cytokines, and chemokines is the key mechanism of CR against obesity.

Based on molecular docking simulation, we observed that the three major alkaloids of CR manifested high-affinity binding with M1 macrophages, M2 macrophages, and TNF. High-affinity binding suggests that a relatively low dose of CR is adequate to trigger the effect of treatment on obesity. Taken together, the findings of the network pharmacology approach suggest that the combined mechanism of CR against obesity may reduce inflammation by modulating macrophages and their associated TNF and chemokine signaling pathways. Therefore, we performed in vivo experiments using an obese mouse model to confirm whether oral administration of CR alleviated metabolic disorders by regulating inflammation in adipose tissue.

Firstly, treatment with CR improved all the weight-related outcomes and downsized lipid droplets in the liver and adipocytes in the epididymal fat pad. We also observed a decreased fat area in the liver and epididymal fat, which implies that CR might act as a repressor of adipogenesis. Zhang et al. reported that berberine, the most abundant compound of CR, suppressed adipogenic genes such as peroxisome proliferator-activated receptor gamma (PPARγ), CCAAT/enhancer binding protein alpha (CEBP/α), and CEBP/β [25]. From these findings, we assumed that the decrease in the fat area by CR administration could have originated from the inhibition of adipogenesis via the modulation of adipogenic transcription factors. Future studies are required to confirm this hypothesis. Previous reports have suggested that weight loss reduces the size of adipocytes and alters their inflammatory and metabolic features, leading to the amelioration of insulin resistance [26]. However, Lifang et al. reported that berberine suppressed proinflammatory response in ATMs without any change in body weight [27]. In several in vitro studies, CR and berberine reduced lipopolysaccharide (LPS)-induced MCP-1/CCL2 production [28], and berberine inhibited MCP-1, IL-6, and TNF-α by downregulating NF-ĸB signaling in murine macrophage cell lines [29]. Therefore, the anti-inflammatory effect of CR seems to be independent of weight loss. Secondly, the CR groups showed a significant improvement in glucose metabolism, as evaluated by OGTT, and lipid metabolism, as evaluated by OFTT. Moreover, the CR 200 group demonstrated beneficial effects on levels of total cholesterol, free fatty acids, HDL cholesterol, and LDL cholesterol. These results corroborate those of previous studies [30,31] and suggest the therapeutic potential of CR in dyslipidemia.

Furthermore, we focused on ATMs from the epididymal fat pad to examine the effects of CR on obesity-induced inflammation. Of the various immune cell types in obese adipose tissue, ATMs are key factors in obesity-linked inflammation, accounting for more than 50% of the immune cells [32]. The elevated number of ATMs in obesity originates from the infiltration of macrophages from monocyte trafficking and the local proliferation of infiltrated macrophages [33]. The recruited macrophages manifest a dramatic change in distribution, developing crown-like structures surrounding the dead adipocytes, and adopting diverse functional properties that are more prone to chronic low-grade inflammation [33]. We examined the infiltration rate of total ATMs and their two major populations, M1 and M2 macrophages, by fluorescence-activated cell sorting. The classically activated M1 macrophages secrete proinflammatory cytokines, such as IL-1β, IL-6, IL-12, IL-23, and TNF-α, in response to stress and infection. In contrast, activated M2 macrophages secrete anti-inflammatory cytokines such as IL-4, IL-10, and TGF-β, maintain homeostasis, and contribute to vasculogenesis and tissue remodeling [34]. Lumeng et al. discovered that ATMs in obese mice are polarized to the M1 state, whereas ATMs in lean mice have an M2 profile [35]. In this study, we observed that the administration of CR led to a significantly decreased percentage of total ATMs that manifested F4/80+ CD45+, suggesting attenuated inflammation. Moreover, both the CR 200 and CR 400 groups showed a decreased percentage of F4/80+ CD45+ CD11c+ M1 macrophages and an increased percentage of F4/80+ CD45+ CD206+ M2 macrophages than that in the control group. Berberine, which is a major alkaloid of CR, has been previously reported to suppress M1 polarization in ATMs [27]. The findings of the present study suggest that the administration of CR as a whole herb reduces chronic low-grade inflammation by decreasing macrophage infiltration and phenotype switching from M1 to M2 macrophages. According to previous studies, the effects of berberine on improving insulin resistance are mainly based on the upregulation of sirtuin 1 (SIRT1) in adipose tissue [36], and its effects on energy metabolism and weight loss are mediated via SIRT1 activation and fibroblast growth factor 21 (FGF21) secretion [37]. Additionally, Xiaoyan et al. reported that adipocyte SIRT1 activation modulated the polarization of ATMs to the anti-inflammatory M2 subset regardless of weight change [38]. Therefore, our findings on the anti-inflammatory actions of CR in adipose tissue seem to be related to SIRT1 activation, which can also explain our results of weight loss through FGF21 downstream [36]. So, further study is necessary to explore the effect of CR on FGF21 or adipose tissue browning.

To investigate the mechanism of the anti-inflammatory effect of CR at the genetic level, we measured the mRNA levels of TNF-α, F4/80, CC chemokines (CCL2, CCL4, and CCL5), and CXC chemokine receptor (CXCR4) in the epididymal fat pad. The CR 200 group showed the most favorable effect of suppressing gene expression in all indices, except CXCR4, whereas the CR 400 group showed significantly reduced expression of TNF-α, F4/80, and CCL5. In the adipose tissue in obese animals, TNF-α is released to a greater extent by infiltrated M1 macrophages and hypertrophied adipocytes [39]. Therefore, a reduced expression of TNF-α indicates the amelioration of adipose tissue inflammation. A decreased expression of F4/80 also implies the downregulation of inflammation, as F4/80 is an established specific cell-surface marker for murine macrophages [40]. We also investigated the effects of CR on chemokines that have pleiotropic roles in inflammation-linked diseases. To date, several chemokines have been identified to induce obesity through the infiltration of immune cells related to obesity-induced inflammation and insulin resistance [41,42,43]. CCL2, also called MCP-1, has a chemotactic ability in monocytes, natural killer cells, memory T cells, and dendritic cells and helps recruit these cells to sites of inflammatory reaction and tissue injury [44]. As CCL2 is highly produced in various diseases related to macrophage infiltration and chronic inflammation [45], the decreased mRNA levels in the CR group indicate a reduction in the inflammatory state. CCL4, also referred to as macrophage inflammatory protein-1β (MIP-1β), is considered a major macrophage attractant that causes obesity-induced chronic inflammation and insulin resistance [46]. CCL5, also known to be regulated on activation, normal T-cell expressed, and secreted (RANTES), increases in the adipose tissue in murine and human obesity, and adipose CCL5 expression was comparatively higher in obese individuals than in lean individuals [47]. Therefore, a reduced expression of CCL4 and CCL5 after administration of CR may decrease inflammation and ameliorate insulin resistance. CXCR4, which is expressed on macrophages and adipocytes in adipose tissue, is a chemokine receptor specific to the stromal-derived-factor-1 (SDF-1 or CXCL12), a strong chemotactic molecule for lymphocytes. The CXCL12-CXCR4 pathway affects adipose tissue by exerting different effects on white or brown adipocytes [48]. In obesity, levels of serum CXCL12 are elevated, and the CXCL12-CXCR4 pathway activates thermogenesis in brown adipose tissue; however, the increased effects of this pathway on white adipose tissue are greater, eventually resulting in insulin resistance [48]. A previous study suggests that CXCL12 is normally produced from various neurons and cells [49], but in obesity, CXCL12 is highly produced from white adipocytes and induces macrophage infiltration in white adipose tissue [50]. We observed a decreased expression of CXCR4 in the CR group than that in the control group, but these differences were not statistically significant.

The potential limitation of this study is that we did not find the optimal dose of CR administration. The outcomes obtained from the CR 200 and CR 400 groups are primarily dose independent; therefore, further study is required to determine the adequate dose for the treatment of obesity. In summary, the study findings indicate that the administration of CR suppresses the gene expression of representative inflammatory cytokines, macrophage markers, and chemokines in obesity.

## 4. Materials and Methods

### 4.1. Network Pharmacology-Based Approach for the Potential Actions of CR on Obesity

#### 4.1.1. Collection of Bioactive Compounds of CR and Their Target Genes

We collected the CR compounds from the TCMSP. After the initial search, the results were filtered with limiting parameters (oral bioavailability (OB) ≥ 30%, Caco2-permeability ≥ 0, and drug-likeness (DL) ≥ 0.18) to obtain bioactive ingredients of CR. Finally, we collected target proteins of each qualified ingredient from SwissTargetPrediction: http://www.swisstargetprediction.ch/ (accessed on 2 January 2021) based on SMILES and excluded some data with a probability of 0.

#### 4.1.2. Search of Obesity-Related Target Genes and Overlapping Genes with CR

We screened the target genes of obesity from three databases (GeneCards, DisGeNET, and OMIM) using the search term “obesity.” We collected disease targets with relevance scores >10 from GeneCards and combined them with targets obtained from the other two databases. Finally, we obtained overlapping genes from obesity and CR and used VENNY 2.1 (https://bioinfogp.cnb.csic.es/tools/venny/) (accessed on 3 January 2021) to express the results using a Venn diagram.

#### 4.1.3. Network Construction of Ingredient–Target and Protein–Protein Interaction

We used Cytoscape 3.7.1 (https://cytoscape.org/) (accessed on 5 January 2021) software to build an interaction network of active ingredient–target protein interactions. In the network, the software defines “node,” which reflects active ingredients of CR and target proteins of obesity, and “edge,” which reflects the relationship between the nodes. We also utilized the STRING platform (https://string-db.org/) (accessed on 5 January 2021) to establish protein–protein interactions (PPIs). The minimum interaction threshold was set to the “highest confidence” (>0.9). “Node” represents the individual targets, and “edge” represents the relationship between the different targets. Finally, we analyzed PPI using Cytoscape and extracted the top 10 target proteins with the highest degrees.

#### 4.1.4. Functional Enrichment Analysis

We collected the GO and KEGG data from the Database for Annotation, Visualization, and Integrated Discovery (DAVID, https://david.ncifcrf.gov/) (accessed on 7 January 2021) and analyzed them using R software to determine the role of target proteins. The results are presented as dot plots showing enrichment terms involved in the molecular function (MF), cellular components (CCs), biological process (BP), and pathways.

#### 4.1.5. Molecular Docking

We explored the interactions between bioactive ingredients and their target proteins to evaluate their binding site and affinity. We selected three major alkaloids of CR with the highest content [51] as ligands and M1 and M2 macrophages as receptors to explore their actions on the macrophages. We also included TNF as a receptor, which is involved in the hub signaling pathways obtained from the functional enrichment analysis. Three-dimensional (3D) structural data of berberine (CID 2353), palmatine (CID 19009), and coptisine (CID 72322) were obtained from PubChem (https://pubchem.ncbi.nlm.nih.gov/) (accessed on 10 January 2021). PDB data (https://www.rcsb.org) (accessed on 10 January 2021) were used to obtain the structures of M1 macrophages (PDBID: 1GD0), M2 macrophages (PDBID: 1JIZ), and TNF (PDBID: 2AZ5). We used the Biovia Discovery Studio Visualizer to pretreat target proteins by removing HETATM and water and adding polar groups. We used Pyrx to perform molecular docking and assess binding affinity and utilized the Biovia Discovery Studio Visualizer to visualize the binding structures. Virtual screening and calculation of the binding score (kcal/mol) were performed using Autodock VINA [52].

### 4.2. In Vivo Experiments to Validate the Potential Actions of CR on Obesity

#### 4.2.1. Preparation of CR and Metformin

CR was provided by the Department of Pharmaceutical Preparation of the Hospital of Korean Medicine, Kyung Hee University (Seoul, Korea). We boiled 100 g of CR with 1500 mL of 80% ethanol for 2 h in a heating mantle. The extract was then transferred to a 500 mL flask for filtration. Next, the filtrate was concentrated using a rotary evaporator (model NE-1, EYELA Co., Tokyo, Japan), and the extract was freeze-dried and stored at room temperature. The final extraction yield of CR was 19.8%. Metformin was obtained from Sigma-Aldrich (St. Louis, MO, USA).

#### 4.2.2. Animals and Diets

Twenty-five 6-week-old male C57BL/6 mice weighing 19–21 g were provided by Central Lab Animals, Inc. The mice were maintained under controlled conditions, with LED lighting (12 h day and 12 h night) and humidity of 40–70%. Water and food were supplied as required. After a week of adaptation period, we distributed 25 mice into five groups (5 mice per group): normal chow (NC), control (HFD), HFD + CR 200 mg/kg (CR 200), HFD + CR 400 mg/kg (CR 400), and HFD + metformin 200 mg/kg (MET) groups and started feeding for 16 weeks. All mice in the control, CR 200, CR 400, and MET groups, except those in the NC group, were fed with HFD containing 60% fats (D12492, Research Diets, New Brunswick, USA).

#### 4.2.3. Drug Administration

After 6 weeks of feeding, when NC and the other 4 groups showed significant differences in the body weight (*p* < 0.001), we orally administered CR 200 mg/kg and 400 mg/kg once daily using zonde for 10 weeks in mice in the CR 200 and CR 400 groups, respectively, while mice in the control group were administered with normal saline. The MET group was orally administered metformin 200 mg/kg once daily during the same period.

#### 4.2.4. Assessment of Weight-Related Outcomes

Body weight (BW) was measured using an electronic scale (CAS 2.5D, Yangjoo, Korea) at the start and end of the study. All mice were assessed at the same time before morning feeding. The mice were placed in a plastic bowl, and we waited until the mice were still in order to minimize the error in measurement caused by their movements. After the mice were terminated at Week 16, we weighed the liver and epididymal fat pad using the electronic scale.

#### 4.2.5. Measurement of Feed Intake

The amount of feed intake in each group was calculated by averaging the daily feed intake. Daily feed intake was determined by measuring the difference in weight between the feed provided on prior day and the feed remaining on the next morning. Measurements were taken every morning before morning feeding. We calculated the caloric intake by multiplying 2.91 kcal/g in the NC group and 5.24 kcal/g in the HFD group.

#### 4.2.6. Oral Glucose Tolerance Test (OGTT)

We performed OGTT at Week 14 after the mice were maintained in a clean cage for a fasting period of 14 h. After fasting blood glucose was tested, we orally administered glucose (2 g/kg body weight) dissolved in distilled water and drew blood samples from the tail vein at 0, 30, 60, 90, 120, and 180 min after administration. Glucose levels were measured using a strip-operated blood glucose sensor (Accu-Chek Performa, Roche, Basel, Switzerland). Based on the OGTT graph, the incremental AUC was calculated.

#### 4.2.7. Measurements of Serum Insulin Level and Insulin Resistance

At Week 16, we collected blood from the tail vein of a 14 h fasting mouse to measure serum insulin concentration. Blood was collected in BD Microtainer serum separator tubes and centrifuged at 2000× *g* for 20 min to obtain serum. Serum insulin levels were measured using an ultrasensitive mouse insulin ELISA kit (Crystal Chem Inc., Chicago, USA). The samples and insulin standards were plated into 96-well antibody-coated microplates (5 μL each) and incubated for 2 h at 4 °C. After five washes, anti-insulin enzyme conjugate was added to each well and incubated at room temperature for 30 min. After seven washes, enzyme substrate solution was added to each well and incubated for 40 min. Finally, reaction stop solution was added, and after 10 min, absorbance was measured using an ELISA reader at 450 nm. From the fasting blood glucose (FBG) and insulin concentration, insulin resistance was evaluated using the homeostatic model assessment of insulin resistance (HOMA-IR). The HOMA-IR was calculated using the following equation: HOMA-IR = fasting blood glucose (mg/dL) × fasting insulin (ng/mL) × 0.0717225161669606.

#### 4.2.8. Oral Fat Tolerance Test (OFTT)

We performed the OFTT at Week 15 after the mice were maintained in a clean cage for a fasting period of 14 h. After fasting triglyceride levels were measured, we orally administered olive oil (2 mL/kg body weight) and drew blood from the tail vein at 0, 120, 240, and 360 min after administration. Triglyceride levels were measured using Accutrend Triglyceride Strip on Accutrend Plus (Roche, CA, USA), and the samples were analyzed using the Triglyceride Colorimetric Assay Kit (Cayman, Ann Arbor, MI, USA). Based on the OFTT graph, the incremental AUC was calculated.

#### 4.2.9. Assessment of Serum Lipid Profiles

At Week 16, we obtained blood samples from the hearts of mice and measured levels of total cholesterol, low-density lipoprotein (LDL) cholesterol, high-density lipoprotein (HDL) cholesterol, triglyceride, phospholipid, and free fatty acids using ELISA kits (Cusabio, Houston, TX, USA).

#### 4.2.10. Evaluation of Liver and Kidney Function

At Week 16, we centrifuged the blood samples obtained from the heart at 3000 rpm for 20 min. Supernatants were stored at −40 °C. We measured aspartate aminotransferase (AST), alanine aminotransferase (ALT), and creatinine levels using ELISA kits (Cusabio, USA) to evaluate the liver and kidney function.

#### 4.2.11. Isolation of RNA from Adipose Tissue

At Week 16, the mice were sacrificed, and the epididymal fat pads were obtained. The samples were immediately wrapped in aluminum foil, placed in liquid nitrogen, and stored at −70 °C until RNA extraction. RNA extraction from the epididymal fat pads was performed using a Mini RNA Isolation Kit (ZYMO RESEARCH, Irvine, CA, USA). The defrosted fat pads were pulverized using a homogenizer in tubes containing 300 µL of ZR RNA buffer and centrifuged at 1000 rpm. The supernatant was transferred to a Zymo-Spin III column in a 2 mL collection tube and centrifuged at 2000 rpm for 1 min. After adding 350 µL of RNA wash buffer to the sample, it was centrifuged for 1 min, washed twice, and then transferred into a 1.5 mL collection tube. Further, 50 µL of RNase-free water was added, and the sample was again centrifuged at 1000 rpm. The final RNA extraction samples were stored at −70 °C until further analysis.

#### 4.2.12. Analysis of Inflammatory Gene Expression

We performed a quantitative real-time polymerase chain reaction (qRT-PCR) to evaluate the expression of inflammatory genes, such as TNF-α, F4/80, CCL2, CCL4, CCL5, and CXCR4. Prior to qRT-PCR, complementary DNA (cDNA) was synthesized using an Advantage RT PCR Kit (Clontech, Palo Alto, CA, USA). We extracted 1 μg of RNA from the epididymal fat pad, oligo (dT) and RNase-free H_2_O were mixed, and the mixture was heated at 70 °C for 2 min. Next, we added a 5× reaction buffer, MMLV reverse transcriptase, recombinant RNase inhibitor, and 10 nM dNTP. Further, the mixture was incubated at 42 °C for 60 min and at 94 °C for 5 min. cDNA was obtained through reverse-transcription PCR: dH_2_O, 2× SYBR reaction buffer, and primers were added, and qRT-PCR was performed using 7900HT Fast Real-Time PCR System (Applied Biosystems, Foster City, CA, USA). The following primers were used: TNF-α, 5′-TTCTG TCTAC TGAAC TTCGG GGTGA TCGGT CC-3′, and 5′-GTATG AGATA GCAAA TCGGC TGACG GTGTGGG-3′; F4/80, 5′-CTTTGGCTATGGGCTTCCAGTC-3′, and 5′-GCAAGGAGGACAGAGTTTATCGTG-3′; CCL2, 5′-AGGTCCCTGTCATGCTTCTGG-3′, and 5′-CTGCTGCTGGTGATCCTCTTG-3′; CCL4, 5′-CTCAGCCCTGATGCTTCTCAC-3′, and 5′-AGAGGGGCAGGAAATCTGAAC-3′; CCL5, 5′-TGCCCACGTCAAGGAGTATTTC-3′, and 5′-AACCCACTTCTTCTCTGGGTTG-3′; CXCR4, 5′-TCAGTGGCTGACCTCCTCTT-3′, and 5′-CTTGGCCTTTGACTGTTGGT-3′; and glyceraldehyde-3-phosphate dehydrogenase (GAPDH, housekeeping gene), 5′-AGTCCATGCCATCACTGCCACC-3′, and 5′-CCAGTGAGCTTCCCGTTCAGC-3′. To analyze gene expression, we converted the threshold cycle (Ct) of each gene obtained by SDS Software 2.4 (Applied Biosystems^®^, USA) to relative quantitation (RQ) with respect to values for GAPDH and calculated the fold change. The fold change value of the experimental group was adjusted with respect to that for the NC group that was considered to be 1.

#### 4.2.13. Isolation of Stromal Vascular Cells (SVCs)

At Week 16, the mice were sacrificed, and adipose tissue samples were obtained from the epididymal fat pad. The samples in solution of phosphate-buffered saline (BSA; Gibco, Waltham, MA, USA) were mixed with 2% bovine serum albumin (BSA; Gibco, Waltham, MA, USA). The samples were cut into smaller pieces of 1–2 mm in size using a round scissor. Further, deoxyribonuclease I (Roche Diagnostic, Indianapolis, IN, USA) and collagenase (Sigma-Aldrich, St. Louis, MO, USA) were added to the sample pieces, and the mixture was shaken for 20–25 min at 37 °C. Next, 5 mM EDTA was added to the 2% BSA/PBS solution, and the samples were filtered through a 100 µm filter (BD Biosciences, San Jose, CA, USA) to eliminate uncrushed lumps of adipose tissue and were centrifuged for 3 min at 1000 rpm. After the supernatant containing adipocytes was separated using a spoide, the remaining solution, except the pellet at the lower layer, was removed. The pellet was mixed with PBS and 2% fetal bovine serum (FBS; Sigma, St. Louis, MO, USA). We eliminated the unnecessary tissues using a 100 µm cell strainer (BD Bioscience, USA). Finally, the samples were centrifuged for 10 min at 200× *g*, and stromal vascular cells (SVCs) were obtained from the bottom of the tube.

#### 4.2.14. Flow Cytometry Analysis of Adipose Tissue Macrophages (ATMs)

After counting the number of SVCs isolated from the adipose tissue, each sample was modulated to comprise 106 cells using a cellometer (Nexcelom Bioscience LLC, Lawrence, MA, USA). FcBlock (BD Pharmingen, USA) was added to the sample at a ratio of 1:100 and allowed to react for 10 min. Fluorophore-conjugated antibodies were added and incubated in the shading state for 20 min. The antibodies used were as follows: CD45-APC Cy7 (BioLegend, USA), F4/80-APC (BioLegend, USA), CD11c-phycoerythrin (CD11b-PE, BioLegend, USA), and CD206-FITC (Biolgend, USA). After washing with 2% FBS/PBS solution and centrifugation at 1500 rpm, the sample was transferred into a fluorescence-activated cell sorting (FACS) tube and analyzed by FACS Calibur (BD Bioscience, USA). Finally, we assessed the percentage of macrophages with CD45(+) F4/80(+), CD45(+)F4/80(+)CD11c(+), and CD45(+)F4/80(+)CD206(+), using FlowJo (Tree Star, Inc., USA).

#### 4.2.15. Histological Analysis of the Liver and Epididymal Fat Pad

First, we fixed the samples obtained from the liver and epididymal fat pads in 10% neutral-buffered formalin. Then, the samples were immersed in 70%, 80%, 95%, and 100% ethanol for dehydration and embedded in paraffin to prepare paraffin blocks. Each tissue sample was sliced into 4 μm thick pieces with a microtome and placed on a gelatin-coated slide. For tissue staining, the slides were dewaxed in xylene and rehydrated in 100%, 95%, 80%, 70% ethanol, and distilled water. The rehydrated tissues were then stained with hematoxylin and eosin (H&E), and digital images were obtained using a high-resolution camera-mounted optical microscope (Olympus BX-50, Olympus Optical, Tokyo, Japan) connected to a computer. Finally, we measured the fat area using ImageJ, which is an open-source software for image analysis.

#### 4.2.16. Statistical Analysis

The experimental data are presented as the mean ± standard error of the mean (SEM), and statistical analysis was performed using GraphPad PRISM 5 (GraphPad Software Inc., San Diego, CA, USA). One-way analysis of variance (ANOVA) was used to evaluate the differences between the groups, followed by Tukey’s post-hoc test. A two-tailed value of *p* < 0.05 was considered statistically significant.

## 5. Conclusions

This study demonstrates that CR is effective in treating obesity-induced inflammation by modulating macrophages and expression of genes related to inflammation in adipose tissue. Therefore, we suggest CR as a promising therapeutic agent for treating obesity and related metabolic disorders.

## Figures and Tables

**Figure 1 ijms-22-08075-f001:**
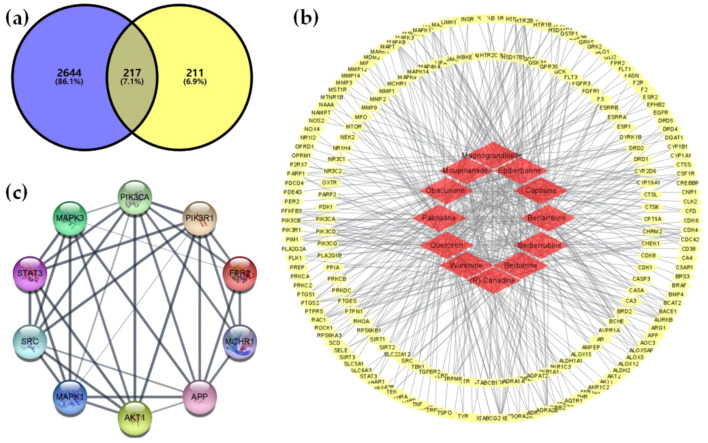
Target genes of CR and obesity and their network. (**a**) The 217 overlapping genes between CR and obesity, (**b**) ingredient–target network, (**c**) and top 10 hub genes with the highest degree of connectivity from the protein–protein interaction analysis.

**Figure 2 ijms-22-08075-f002:**
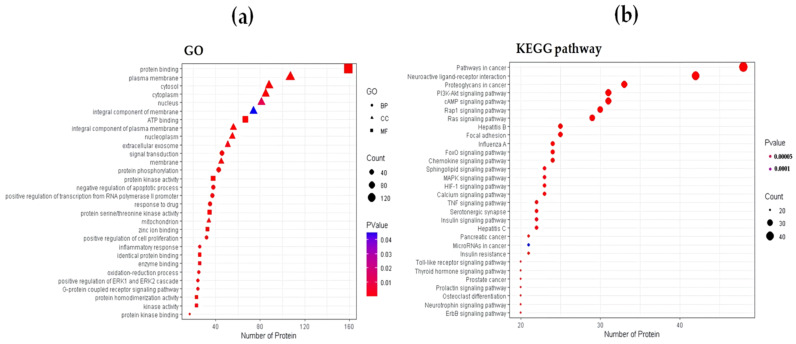
Functional enrichment analysis. (**a**) GO enrichment analysis and (**b**) KEGG pathway enrichment analysis.

**Figure 3 ijms-22-08075-f003:**
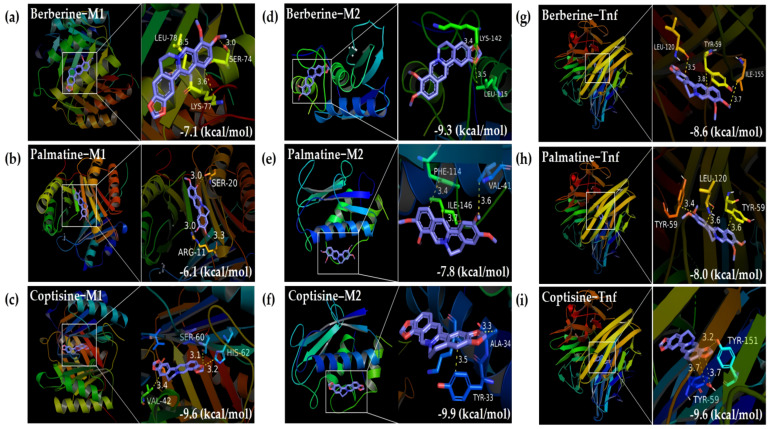
Molecular docking between the representative components of CR and M1, M2 macrophages, and TNF. (**a**) Berberine and M1 macrophage, (**b**) berberine and M2 macrophage, (**c**) palmatine and M1 macrophage, (**d**) palmatine and M2 macrophage, (**e**) coptisine and M1 macrophage, (**f**) coptisine and M2 macrophage, (**g**) berberine and TNF, (**h**) palmatine and TNF, and (**i**) coptisine and TNF. M1 macrophage (PDBID: 1GD0), M2 macrophage (PDBID: 1JIZ), and TNF (PDBID: 2AZ5). Binding scores are marked on the bottom right of each subpanel.

**Figure 4 ijms-22-08075-f004:**
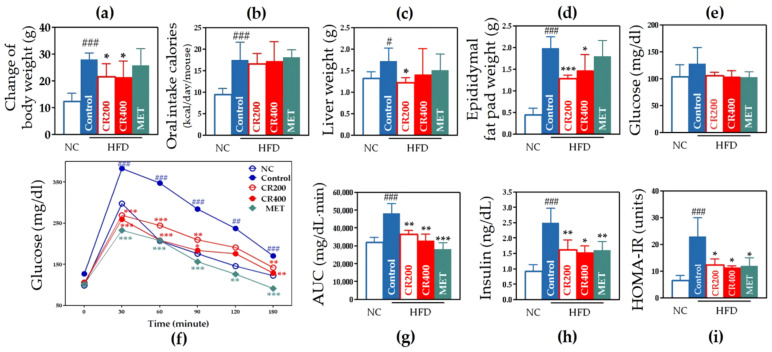
Weight- and glucose-metabolism-related outcomes. (**a**) Change in body weight, (**b**) caloric intake, (**c**) liver weight, (**d**) epididymal fat pad weight, (**e**) fasting glucose level, (**f**) OGTT, (**g**) AUC of OGTT, (**h**) serum insulin level, and (**i**) HOMA-IR. Data are expressed as the mean ± standard error of the mean (SEM). # *p* < 0.05, ## *p* < 0.01, ### *p* < 0.001, control compared with NC, and * *p* < 0.05, ** *p* < 0.01, *** *p* < 0.001, CR 200, CR 400, and MET compared with control. AUC, area under the curve; NC, normal chow; HFD, high-fat diet; MET, metformin; HOMA-IR, homeostatic model assessment of insulin resistance; OGTT, oral glucose tolerance test.

**Figure 5 ijms-22-08075-f005:**
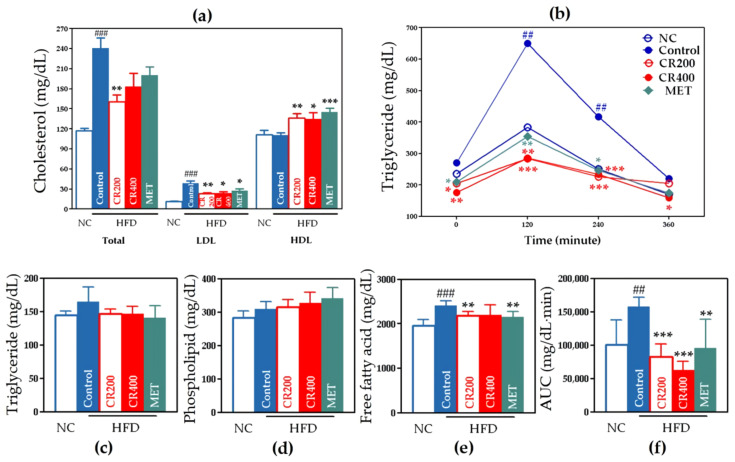
Lipid-metabolism-related outcomes. (**a**) Total, LDL and HDL cholesterol, (**b**) OFTT, (**c**) triglyceride, (**d**) phospholipid, (**e**) free fatty acid, and (**f**) AUC of OFTT. Data are expressed as the mean ± standard error of the mean (SEM). ## *p* < 0.01, ### *p* < 0.001, control compared with NC, and * *p* < 0.05, ** *p* < 0.01, *** *p* < 0.001, CR 200, CR 400, and MET compared with control. AUC, area under the curve; NC, normal chow; HFD, high-fat diet; MET, metformin; OFTT, oral fat tolerance test.

**Figure 6 ijms-22-08075-f006:**
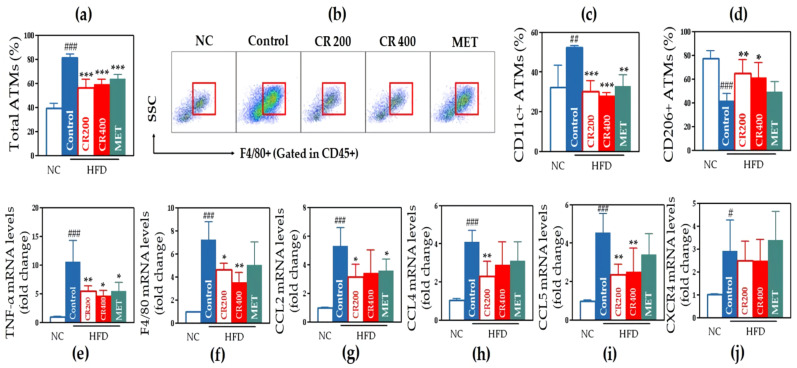
Analysis of ATMs and gene expression of inflammatory markers in adipose tissue. (**a**) Percentage of total ATMs, (**b**) flow cytometry of total ATMs, (**c**) percentage of CD11c+ ATMs, (**d**) percentage of CD206+ ATMs, and expression of (**e**) TNF-α, (**f**) F4/80, (**g**) CCL2, (**h**) CCL4, (**i**) CCL5, and (**j**) CXCR4. Data are expressed as the mean ± standard error of the mean (SEM). # *p* < 0.05, ## *p* < 0.01, ### *p* < 0.001, control compared with NC, and * *p* < 0.05, ** *p* < 0.01, *** *p* < 0.001, CR 200, CR 400 and MET compared with control. ATMs, adipose tissue macrophages; NC, normal chow; HFD, high-fat diet; MET, metformin.

**Figure 7 ijms-22-08075-f007:**
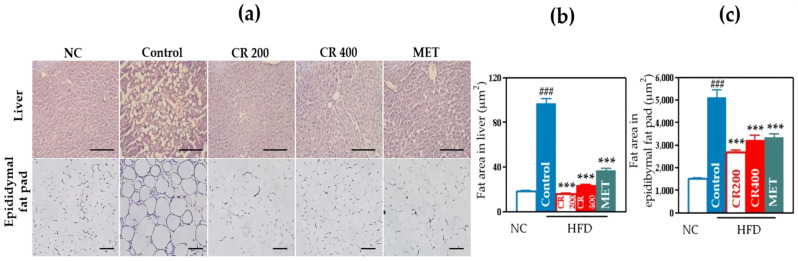
Histological analysis. (**a**) Representative histological images, scale bar indicates 100 µm, (**b**) fat area in the liver, and (**c**) fat area in the epididymal fat pad. Data are expressed as the mean ± standard error of the mean (SEM). ### *p* < 0.001, control compared with NC, and *** *p* < 0.001, CR 200, CR 400 and MET compared with control. NC, normal chow; HFD, high-fat diet; MET, metformin.

## Data Availability

The data presented in this study are available in Appendix A.

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
