# Peer review of "Mechanisms and Effect of Coptidis Rhizoma on Obesity-Induced Inflammation: In Silico and In Vivo Approaches"

_ijms, 2021, doi:10.3390/ijms22158075_

Round 1
Reviewer 1 Report
This manuscript titled „Mechanisms and Effect of Coptidis Rhizoma on Obesity-Induced Inflammation: in silico and in vivo Approaches“, written by OJ Kwon et al, consists of two parts of study: (i) in silico and (ii) in vivo. Although the combination of in silico study, focused on the potential actions of Coptidis Rhizoma (CR) on obesity and related inflammation, and in vivo study, verifying CR effects in mice model, is very interesting, I think, that the potential of this combination study was not fully utilize.
The results of in silico analysis is insufficiently described in contrast to Discussion, where the individual pathways leading to obesity and an inflammation are well characterized. I would recommend focusing on better defining the results of the in silico study and on the links within the in silico and in vivo studies.
In the in vivo study, the metformin treated group with was included without any introduction and explanation of the results and discussion, this group appears to be redundant in this study. Based on this finding, the metformin group could be omitted or introduced and the results should be discussed. How did you determine the dose of metformin?
According to the description of the Materials and Methods, the in vivo study was based on reversal study because mice were fed a high-fat diet for 6 weeks (except for one group fed chow) and then treated daily with CR or metformin. In the results should be specify that this is a reversal study and body weight data at the time of initiation of treatment, i.e. after 6 weeks of a high fat diet, should be added to be ensure that body weight did not differ among groups.
Although the experiments are well designed and well described, I have several comments.
- The Introduction is well written, but I would appreciate more details about the involvement of CR in the study of obesity and related inflammation. The goal of the study should be more specified.
- Results:
The position of the picture description (a), (b) ... is indistinct.
I recommend changing the colors of the individual bars in the charts, especially the better color resolution of colors between control and metformin groups.
2.1. Network pharmacology-based analysis:
Material and methods are described at the end of manuscript and due to I would like to point out the abbreviations that are not introduced in time.
In results, I would appreciate broader description and specifications in obtaining the results and why they are important for further analysis. The interconnection among individual analyses in silico study and especially between in silico study and in vivo study is insufficiently.
2.2. Evaluation of anti-obesity effects of CR in vivo:
An introduction to the in vivo experiment should be added at the beginning of the Results.
Description: We observed a significant weight gain .... (line 103) is not correct due to the measurement of body weight at the end of the experiment, but not the change in body weight between the beginning and the end or during the whole experiment.
Calculation of HOMA-IR – insulin levels were measured after 6 hours fasting in contrast to glucose levels, measured after 14 hours of fasting. Both insulin and glucose should be measured at the same conditions.
Figure 4 (f) – specification of AUC is missing, did you calculate total or incremental AUC? I recommend to add chart of fasting glucose.
Figure 4 (g) – description of units missing in chart
Could you comment on any differences in plasma levels of triacylglycerol between chow and high fat groups? Higher triacylglycerol levels in the high fat group could be expected.
2.2.5. Effects on adipose tissue macrophages (ATMs)
I recommend better characterization of individual markers obtained from FACS measurement and interconnection with Discussion. Markers M1 and M2 are used only in Discussion and in results are missing. An unification is necessary for better understanding.
2.2.6. Effects on inflammatory gene expression in the adipose tissue
As written above, better characterization of M1 and M2 (inflammatory vs anti-inflammatory) should be added. Measurement of genes of anti-inflammatory markers (Arg1, ...) should be add. I also recommend to measuring of the genes from pathways, mentioned by in the in silico analysis associated with obesity and inflammation observed in in vivo experiments.
Figure 7. Higher resolution of the images of immunohistology analysis
- Materials and Methods
4.2.2 Specification of the diet should be added. Number of animals per group and repeating of experiment should be added.
4.2.6. Measurement of basal glucose and 180 min is missing in description
4.2.6. and 4.2.8. calculation AUC should be added, type of AUC (total or incremental) should be specify
4.2.9. and 4.2.10. Methods of measurement should be specified
Supplementary Materials
Supplementary Figure S2: Statistics and abbreviations should be added
Supplementary Figure S2: abbreviations should be added
Author Response
Answers to the Reviewer
Manuscript No.: ijms- 1298516
Authors: Oh-Jun Kwonet al.
Title: “Mechanisms and Effect of Coptidis Rhizoma on Obesity-Induced Inflammation: in silico and in vivo Approaches”
Thank you very much for considering our manuscript for publication. Your suggestions were very helpful to us, and we have incorporated those points into our revised manuscript.
The changes made to the manuscript are as follows:
Reviewer 1
- This manuscript titled „Mechanisms and Effect of Coptidis Rhizoma on Obesity-Induced Inflammation: in silicoand in vivoApproaches“, written by OJ Kwon et al, consists of two parts of study: (i) in silico and (ii) in vivo. Although the combination of in silico study, focused on the potential actions of Coptidis Rhizoma (CR) on obesity and related inflammation, and in vivo study, verifying CR effects in mice model, is very interesting, I think, that the potential of this combination study was not fully utilize.
- The results of in silico analysis is insufficiently described in contrast to Discussion, where the individual pathways leading to obesity and an inflammation are well characterized. I would recommend focusing on better defining the results of the in silico study and on the links within the in silicoand in vivo
▶ As your comment, we added the explanation in the results section on in silico analysis and its links to in vivo experiment.
New Manuscript : Page 2. Results 2.1.2 Analysis of protein-protein interaction (PPI), line 2
These 10 hub genes, which are described in detail in the discussion, are mainly associated with macrophage activation in obesity.
New Manuscript : Page 3. Results 2.1.3 Functional enrichment analysis, line 21
Among those obtained pathways, we noticed TNF signaling pathway and chemokine signaling pathway which are closely related to macrophage activation in adipose tissue. These results provided clues about the mechanisms of CR in treating obesity and the design of in vivo experiment to confirm its effects.
New Manuscript : Page 4. Results 2.1.4 Molecular docking, line 1
Molecular docking was the last in silico approach before the initiation of in vivo experiment. We aimed to simulate whether CR could trigger sufficient effects on macrophages and their released TNF based on our earlier in silico findings.
- In the in vivostudy, the metformin treated group with was included without any introduction and explanation of the results and discussion, this group appears to be redundant in this study. Based on this finding, the metformin group could be omitted or introduced and the results should be discussed. How did you determine the dose of metformin?
▶ As your comment, we added the explanation on metformin group which was positive control in our study. We determined the dose of metformin according to previous animal studies administering 200mg/kg/day of metformin. (Zhang, X.; Zhao, Y.; Xu, J.; Xue, Z.; Zhang, M.; Pang, X.; Zhang, X.; Zhao, L., Modulation of gut microbiota by berberine and metformin during the treatment of high-fat diet-induced obesity in rats. Sci Rep 2015, 5, 14405)
New Manuscript : Page 1. Introduction. paragraph 2, line 11
Metformin, which is first-line medication for type 2 diabetes, was used as positive control since it has been reported to improve glucose intolerance and reduce body weight gain in high-fat diet-fed mice [8]
- According to the description of the Materials and Methods, the in vivostudy was based on reversal study because mice were fed a high-fat diet for 6 weeks (except for one group fed chow) and then treated daily with CR or metformin. In the results should be specify that this is a reversal study and body weight data at the time of initiation of treatment, i.e. after 6 weeks of a high fat diet, should be added to be ensure that body weight did not differ among groups.
▶ We provided normal chow or high-fat diet to each group for 16 weeks and initiated drug administration on 7th week when we observed significant weight difference between groups. We added the body weight changes of all experimental period in Supplementary Figure 2S.
- Although the experiments are well designed and well described, I have several comments.
- The Introduction is well written, but I would appreciate more details about the involvement of CR in the study of obesity and related inflammation. The goal of the study should be more specified.
▶ As your comment, we revised the phrases and added explanation on the goal of this study in the introduction.
New Manuscript : Page 2. Introduction, paragraph 2, line 6
However, most previous studies have focused on actions of specific alkaloids of CR such as berberine even though CR is prescribed as whole herb in oriental medicine. Hence, the goal of this study is to screen potential mechanisms of CR itself on obesity and investigate its effect by evaluating phenotypic changes, ATMs populations and gene expression of inflammatory markers.
- Results:
The position of the picture description (a), (b) ... is indistinct.
▶ We changed the positions of Figures descriptions as your comment.
Figure 1, Figure 2, Figure 4, Figure 5, Figure 6 and Figure7.
- I recommend changing the colors of the individual bars in the charts, especially the better color resolution of colors between control and metformin groups.
▶ We changed the color settings according to your comment.
- 1. Network pharmacology-based analysis:
Material and methods are described at the end of manuscript and due to I would like to point out the abbreviations that are not introduced in time.
▶ We supplemented the phrases to introduce the abbreviations in time.
New Manuscript : page 3, paragraph 1, line 8-10
phosphatidylinositol3‑kinase (PI3K)/protein kinase B (AKT) signaling pathway
cyclic adenosine monophosphate (cAMP) signaling pathway
mitogen-activated protein kinase (MAPK) signaling pathway
- In results, I would appreciate broader description and specifications in obtaining the results and why they are important for further analysis. The interconnection among individual analyses in silicostudy and especially between in silicostudy and in vivo study is insufficiently.
▶ As your comment, we added the explanation in the results section on in silico analysis and its links to in vivo experiment.
New Manuscript : Page 2. Results 2.1.2 Analysis of protein-protein interaction (PPI), line 2
These 10 hub genes, which are described in detail in the discussion, are mainly associated with macrophage activation in obesity.
New Manuscript : Page 3. Results 2.1.3 Functional enrichment analysis, line 21
Among those obtained pathways, we noticed TNF signaling pathway and chemokine signaling pathway which are closely related to macrophage activation in adipose tissue. These results provided clues about the mechanisms of CR in treating obesity and the design of in vivo experiment to confirm its effects.
New Manuscript : Page 4. Results 2.1.4 Molecular docking, line 1
Molecular docking was the last in silico approach before the initiation of in vivo experiment. We aimed to simulate whether CR could trigger sufficient effects on macrophages and their released TNF based on our earlier in silico findings.
- 2. Evaluation of anti-obesity effects of CR in vivo:
An introduction to the in vivo experiment should be added at the beginning of the Results.
▶ We added the explanation at the beginning of the results.
New Manuscript : Page 4, Results 2.2. Evaluation of anti-obesity effects of CR in vivo, line 1
Through in silico screening, we have found that CR might act on macrophage activation and modulate inflammation in treating obesity. Thus, we conducted in vivo experiment to confirm its anti-inflammatory effects and discovered that CR regulated macrophage infiltration and gene expression of inflammatory markers. We have pointed out macrophage-mediated inflammation as the key factor connecting in silico and in vivo approaches.
- Description: We observed a significant weight gain .... (line 103) is not correct due to the measurement of body weight at the end of the experiment, but not the change in body weight between the beginning and the end or during the whole experiment.
▶ We added the body weight changes of all experimental period in Supplementary Figure 2S.
- Calculation of HOMA-IR – insulin levels were measured after 6 hours fasting in contrast to glucose levels, measured after 14 hours of fasting. Both insulin and glucose should be measured at the same conditions.
▶ We corrected the time 6hour to 14hour of when fasting insulin and HOMA-IR were calculated.
- Figure 4 (f) – specification of AUC is missing, did you calculate total or incremental AUC? I recommend to add chart of fasting glucose.
▶ We calculated incremental AUC. We added chart of fasting glucose in figure 4e.
- Figure 4 (g) – description of units missing in chart
▶ We corrected it.
- Could you comment on any differences in plasma levels of triacylglycerol between chow and high fat groups? Higher triacylglycerol levels in the high fat group could be expected.
▶ We observed higher triglyceride in high fat diet group compared to normal chow group, but the difference was not statistically significant. It might be due to the small number of animal models.
- 2.5. Effects on adipose tissue macrophages (ATMs)
I recommend better characterization of individual markers obtained from FACS measurement and interconnection with Discussion. Markers M1 and M2 are used only in Discussion and in results are missing. An unification is necessary for better understanding.
▶ We added the term M1, M2 macrophages in the 2.2.5 result section.
- 2.6. Effects on inflammatory gene expression in the adipose tissue
As written above, better characterization of M1 and M2 (inflammatory vs anti-inflammatory) should be added. Measurement of genes of anti-inflammatory markers (Arg1, ...) should be add. I also recommend to measuring of the genes from pathways, mentioned by in the in silico analysis associated with obesity and inflammation observed in in vivo experiments.
▶ We aggregated findings from in silico approach and focused especially on macrophage activation, so we carried out experiments discovering effects of CR administration on adipose tissue macrophages. We agree with your comment that it would have been better if we investigated other inflammatory markers or genes from pathways, so further studies including those points are necessary to provide more robust evidence.
- Figure 7. Higher resolution of the images of immunohistology analysis
▶ We exchanged the image with higher resolution.
- Materials and Methods
4.2.2 Specification of the diet should be added. Number of animals per group and repeating of experiment should be added.
▶ We provided more information on the diet in the 4.2.2 section. Number of animals per group is also added.
After a week of adaptation period, we distributed 25 mice into five groups (5 mice per group): normal chow (NC), control (HFD), HFD + CR 200 mg/kg (CR 200), HFD + CR 400 mg/kg (CR 400), and HFD + metformin 200 mg/kg (MET) groups and started feeding for 16 weeks. All mice in the control, CR 200, CR 400, and MET groups, except those in the NC group, were fed with HFD containing 60% fats (D12492, Research diets).
- 2.6. Measurement of basal glucose and 180 min is missing in description
▶ We corrected that part in 4.2.6.
- 2.6. and 4.2.8. calculation AUC should be added, type of AUC (total or incremental) should be specify
▶ We added that information as your comment.
4.2.6. Based on OGTT graph, incremental AUC was calculated.
4.2.10. Based on OFTT graph, incremental AUC was calculated.
- 2.9. and 4.2.10. Methods of measurement should be specified
▶ We supplemented more explanation.
4.2.9. using ELISA kits (Cusabio, USA).
4.2.10. using ELISA kits (Cusabio, USA)
- Supplementary Materials
Supplementary Figure S2: Statistics and abbreviations should be added
▶ Statistics and abbreviations were added in Supplementary Figure 2S as your comment.
Supplementary Figure S2: abbreviations should be added
▶ Abbreviations were added in Supplementary Figure 2S as your comment.
Reviewer 2 Report
Kwon et al. aimed to investigate the effects of Coptidis Rhizoma (CR) on obesity-induced (adipose tissue) inflammation. Based on in silico approaches, they suggest that CR may reduce inflammation and ameliorate insulin resistance in obesity. Moreover, they find that several bioactive ingredients of CR show high-affinity binding to macrophages and TNFα. In vivo, they show that CR treatment reduces HFD-induced obesity, glucose intolerance, insulin resistance and dyslipidaemia. Flow cytometry analysis revealed that CR treatment reduced macrophage infiltration and changed the polarization spectrum from M1 to M2 macrophages in white adipose tissue of HFD-fed mice. Moreover, mRNA expression of pro-inflammatory markers was reduced, further suggesting that CR reduced macrophage infiltration/M1 polarization. In addition, histological assessment suggested decreased lipid accumulation in livers as well as decreased adipocyte size in CR-treated HFD-fed mice. They conclude that CR hast a therapeutic effect on obesity-induced adipose tissue inflammation as it supresses inflammatory cytokine production form adipose tissue macrophages.
Broad comments:
Confirming previous findings, authors show that CR reduces body weight gain in HFD-fed mice. As blunted weight gain is known to reduce adipocytes size and WAT inflammation, it is important to clarify the reason for the reduced body weight gain in CR-treated mice. In the Materials and Methods section, authors mention that they measured food intake, however, no such data is reported. Besides food intake, authors need to determine energy expenditure in CR treated mice. Of note, the main active CR compound berberine was shown to induce energy expenditure (Br J Pharmacol 2018;175(2):374-387). Hence, reduced WAT inflammation may indirectly result from reduced weight gain rather than from a direct effect of CR on WAT inflammation.
Specific comments:
In order to investigate whether CR affects WAT inflammation independent of the effect on body weight, WAT inflammation needs to be assessed in CR treated mice at a time point where body weight does not yet differ. Along the same line, authors should show body weight curves over time.
It has also been shown that CR inhibits adipogenesis (Fitoterapia 2014; 98:199-208). Were PPARg and CEBP/b levels decreased in adipose tissue of CR treated mice?
The CR ingredient berberine decreases M1 macrophage activation in adipose tissue and improves insulin sensitivity (Life Sci 2016;166:82-91)). This paper should be discussed.
Figures are not always presented in a numerical order (e.g. Figure 1c appears before Figure 1b in the text, Figure 4g before 4 d, e). Please adapt accordingly.
Authors present some data (e.g. Figure 4 and 6) in Figures and also mention corresponding numbers in the text. However, data should only be presented once (either in a Figure or as numbers in the text). Please adapt accordingly.
Line 182: authors refer to adipocytes in liver. Are these indeed adipocytes or rather lipid droplet in hepatocytes?
Please state the catalogue number of the used HFD in the Materials and Methods section.
Author Response
Answers to the Reviewer
Manuscript No.: ijms- 1298516
Authors: Oh-Jun Kwonet al.
Title: “Mechanisms and Effect of Coptidis Rhizoma on Obesity-Induced Inflammation: in silico and in vivo Approaches”
Thank you very much for considering our manuscript for publication. Your suggestions were very helpful to us, and we have incorporated those points into our revised manuscript.
The changes made to the manuscript are as follows:
Reviewer 2
Kwon et al. aimed to investigate the effects of Coptidis Rhizoma (CR) on obesity-induced (adipose tissue) inflammation. Based on in silico approaches, they suggest that CR may reduce inflammation and ameliorate insulin resistance in obesity. Moreover, they find that several bioactive ingredients of CR show high-affinity binding to macrophages and TNFα. In vivo, they show that CR treatment reduces HFD-induced obesity, glucose intolerance, insulin resistance and dyslipidaemia. Flow cytometry analysis revealed that CR treatment reduced macrophage infiltration and changed the polarization spectrum from M1 to M2 macrophages in white adipose tissue of HFD-fed mice. Moreover, mRNA expression of pro-inflammatory markers was reduced, further suggesting that CR reduced macrophage infiltration/M1 polarization. In addition, histological assessment suggested decreased lipid accumulation in livers as well as decreased adipocyte size in CR-treated HFD-fed mice. They conclude that CR hast a therapeutic effect on obesity-induced adipose tissue inflammation as it supresses inflammatory cytokine production form adipose tissue macrophages.
Broad comments:
- Confirming previous findings, authors show that CR reduces body weight gain in HFD-fed mice. As blunted weight gain is known to reduce adipocytes size and WAT inflammation, it is important to clarify the reason for the reduced body weight gain in CR-treated mice. In the Materials and Methodssection, authors mention that they measured food intake, however, no such data is reported. Besides food intake, authors need to determine energy expenditure in CR treated mice. Of note, the main active CR compound berberine was shown to induce energy expenditure (Br J Pharmacol 2018;175(2):374-387). Hence, reduced WAT inflammation may indirectly result from reduced weight gain rather than from a direct effect of CR on WAT inflammation.
▶ We didn’t measure the energy expenditure of experimental mice. However, we confirmed that there was no difference in oral intake calories in each experimental group. We added the graph as figure 4b. (New Manuscript : page 5, figure 4b)
▶ As you commented, Yixuan et al. (Sun, Y.; Xia, M.; Yan, H.; Han, Y.; Zhang, F.; Hu, Z.; Cui, A.; Ma, F.; Liu, Z.; Gong, Q.; Chen, X.; Gao, J.; Bian, H.; Tan, Y.; Li, Y.; Gao, X., Berberine attenuates hepatic steatosis and enhances energy expenditure in mice by inducing autophagy and fibroblast growth factor 21. Br J Pharmacol 2018, 175, (2), 374-387.) suggested that berberine promotes SIRT1 activation and FGF21 secretion, resulting in browning of white adipose tissue (Br J Pharmacol 2018;175(2):374-387). The increased energy expenditure of berberine may also be a possible mechanism for the weight loss of CR. However, our findings showed that CR treatment effects not o[1]nly on weight loss but on improvement in glucose and lipid metabolism, and we focused on the anti-inflammatory mechanisms.
Specific comments:
- In order to investigate whether CR affects WAT inflammation independent of the effect on body weight, WAT inflammation needs to be assessed in CR treated mice at a time point where body weight does not yet differ. Along the same line, authors should show body weight curves over time.
▶ We did not assess WAT inflammation at a time point without weight difference.
▶ In our findings, CR treated mice showed not only weight loss, but anti-inflammatory responses. In previous studies, an in vivo study (Life Sci 2016;166:82-91) reported berberine suppresses WAT inflammation without weight change and several in vitro studies with RAW164.7 cell line suggested that CR and berberine inhibit pro-inflammatory cytokines by downregulating NF-ĸB signaling via SIRT1 pathway. Taking together, the anti-inflammatory effect of CR seems to be independent to the weight loss. We discussed this point in discussion section as below.
New Manuscript : page 9, paragraph 3, line 4
However, Lifang et al. reported that berberine suppresses pro-inflammatory response in ATMs without any change in body weight [26]. In several in vitro studies, CR and berberine reduce LPS-induced MCP-1/CCL2 production [27], and berberine inhibits MCP-1, IL-6 and TNF-α downregulating NF-ĸB signaling via SIRT1 pathway in murine macrophage cell lines [28]. Therefore, the anti-inflammatory effect of CR seems to be independent to the weight loss.
- It has also been shown that CR inhibits adipogenesis (Fitoterapia 2014; 98:199-208). Were PPARg and CEBP/b levels decreased in adipose tissue of CR treated mice?
▶ As you commented, CR has adipogenesis inhibitory effect, but we focused on the anti-inflammatory effect and its metabolic improvement. So, we didn’t examine the levels of PPARg and CEBP/b in adipose tissue of CR treated mice. I would like to ask your understanding about it.
- The CR ingredient berberine decreases M1 macrophage activation in adipose tissue and improves insulin sensitivity (Life Sci 2016;166:82-91)). This paper should be discussed.
▶ As you commented, we added the reference and discussed along with the result of CR on ATMs in the discussion section as below.
New Manuscript : page 9, paragraph 4, line 21
Berberine, which is a major alkaloid of CR, has been previously reported to suppress M1 polarization in ATMs [26].
- Figures are not always presented in a numerical order (e.g. Figure 1c appears before Figure 1b in the text, Figure 4g before 4 d, e). Please adapt accordingly.
▶ We adapted figures in numerical order.
- Authors present some data (e.g. Figure 4 and 6) in Figures and also mention corresponding numbers in the text. However, data should only be presented once (either in a Figure or as numbers in the text). Please adapt accordingly.
▶ Since we presented results by figures, we deleted the numbers in the text as your comment.
- Line 182: authors refer to adipocytes in liver. Are these indeed adipocytes or rather lipid droplet in hepatocytes?
▶ As you pointed out, we observed lipid droplet in hepatocytes while we examined adipocytes in adipose tissue. We corrected as below.
New Manuscript : page 7, 2.2.7 Effects on the size of lipid droplets in hepatic tissue and adipocytes in adipose tissue
We observed that the control group showed a prominent increase in the size of lipid droplets in the liver and adipocytes in the epididymal fat pad than NC group. Both the CR 200 and CR 400 groups showed significantly downsized lipid droplets and adipocytes than HFD group.
- Please state the catalogue number of the used HFD in the Materials and Methods
▶The catalogue number of the used HFD was D12492 from Research diets.
New Manuscript : page 12, paragraph 1, line 4
…were fed with HFD containing 60% fats (D12492, Research diets).
Round 2
Reviewer 2 Report
While the authors have adequately responded to some of my questions, they did not perform any further experiments I was asking for. I expect them to analyse markers of adipogenesis (e.g. PPARg and CEBP/b) as requested, as they have access to RNA isolated from adipose tissue. I still believe that it is valuable to check whether CR affects adipogenesis, although the focus of their study was on WAT inflammation. Similarly, I understand that authors did not measure energy expenditure and it would be time consuming to redo such analysis. However, they can assess some readouts associated with energy expenditure recently shown to be affected by berberine such as FGF21 levels or adipose tissue browning.
Author Response
Answers to the Reviewer
Manuscript No.: ijms- 1298516
Authors: Oh-Jun Kwonet al.
Title: “Mechanisms and Effect of Coptidis Rhizoma on Obesity-Induced Inflammation: in silico and in vivo Approaches”
Thank you very much for considering our manuscript for publication. Your suggestions were very helpful to us, and we have incorporated those points into our revised manuscript.
The changes made to the manuscript are as follows:
Reviewer 2
While the authors have adequately responded to some of my questions, they did not perform any further experiments I was asking for. I expect them to analyse markers of adipogenesis (e.g. PPARg and CEBP/b) as requested, as they have access to RNA isolated from adipose tissue. I still believe that it is valuable to check whether CR affects adipogenesis, although the focus of their study was on WAT inflammation. Similarly, I understand that authors did not measure energy expenditure and it would be time consuming to redo such analysis. However, they can assess some readouts associated with energy expenditure recently shown to be affected by berberine such as FGF21 levels or adipose tissue browning.
- In this animal study, we obtained two epididymal fat pads from each mouse, of which 1/2 epididymal pad were used for FACS, another 1/4 were used for histological analysis, and the other 1/4 were used for gene expression analysis. As you well know, due to high fatty acids content and low cell number in adipose tissue, the isolation of RNA from adipose tissue results in poor yield of RNA.
We fully agree with your comments, so we planned the additional gene expression experiments of PPARg, CEBP/b, FGF21 and other adipose tissue browning-related genes. However, considering the small amount of RNA from 1/4 adipose tissue, the RNA replication process with the remaining RNA samples is required to perform additional RT-PCR, and we do not have enough time to complete the procedures, including purchasing the replication kits and primers, performing RNA replication and RT-PCR, and analyzing the additional gene expressions before the submission due date (25th July) of minor revisions. Instead, we add some discussion about your points in the discussion section as below. We kindly ask for your understanding about our situation again.
New Manuscript : page 8, paragraph 4, line 2
We also observed decreased fat area in liver and epididymal fat which implies that CR might act as repressors of adipogenesis. Zhang et al. have reported that berberine, the most abundant compound of CR, suppressed adipogenic genes such as peroxisome proliferators-activated receptor gamma (PPARγ), CCAAT/enhancer binding protein alpha (CEBP/α) and CEBP/β [25]. From these findings, we assumed that the decrease in fat area by CR administration could be originated from inhibition of adipogenesis via modulation of adipogenic transcription factors. Future studies are required to confirm this hypothesis.
New Manuscript : page 9, paragraph 2, line 25
According to the previous studies, effects of berberine on improving insulin resistance is mainly based on upregulation of sirtuin 1 (SIRT1) in adipose tissue [36], and its effects on energy metabolism and weight loss are mediated via SIRT1 activation and fibroblast growth factor 21 (FGF21) secretion [37]. Additionally, Xiaoyan et al. have reported that adipocyte SIRT1 activation modulated the polarization of ATMs to the anti-inflammatory M2 subset regardless of weight change [38]. Therefore, our findings on anti-inflammatory actions of CR in adipose tissue seems to be related to SIRT1 activation, which can also explain our results of weight loss through FGF21 downstream [36]. So, further study is necessary to explore the effect of CR on FGF21 or adipose tissue browning.
We thank you again for your insightful comments on our paper.
Sincerely yours,
Byung-Cheol Lee, M.D.& Ph.D.
